**Data Availability Statement:** All relevant data are within the paper and its Supporting Information files.

# Clinicians' prescribing pattern, rate of patients' medication adherence and its determinants among adult hypertensive patients at Jimma University Medical Center: Prospective cohort study

**Bekalu Kebede Simegn**[1]*, **Legese Chelkeba**[2], **Bekalu Dessie Alamirew**[1]

1 Department of Pharmacy, College of Health Sciences, Debre Markos University, Debre Markos, Ethiopia,
2 Department of Clinical Pharmacy, School of Pharmacy, College Health Sciences, Addis Ababa University, Addis Ababa, Ethiopia

* bekalukebede19@gmail.com

## Abstract

### Background

Many studies conducted in the past focused on patients' sociodemographic factors and medical profiles to identify the determinants of suboptimal blood pressure control. However, prescribing patterns and clinicians' adherence to guidelines are also important factors affecting the rate of blood pressure control. Therefore, this study aimed to determine clinicians' prescribing patterns, patients' medication adherence, and its determinants among hypertensive patients at Jimma University Medical Center.

### Methods

A general prospective cohort study was conducted among hypertensive patients who had regular follow-up at Jimma university ambulatory cardiac clinic from March 20, 2018, to June 20, 2018. Patients' specific data was collected with a face-to-face interview and from their medical charts. Clinicians' related data were collected through a self-administered questionnaire. Data were analyzed using SPSS version 21.0. Bivariate and multivariable logistic regression analyses were done to identify key independent variables influencing patients' adherence. P-Values of less than 0.05 were considered statically significant.

### Results

From the total of 416 patients, 237(57.0%) of them were males with a mean age of 56.50 ± 11.96 years. Angiotensin-converting enzyme inhibitors were the most frequently prescribed class of antihypertensives, accounting for 261(63.7%) prescriptions. Combination therapy was used by the majority of patients, with 275 (66.1%) patients receiving two or more antihypertensive drugs. Patients' medication adherence was 46.6%, while clinicians' guideline adherence was 44.2%. Patients with merchant occupation (P = 0.020), physical inactivity (P

**Funding:** This research was funded by Jimma University, College of Health Sciences. The University has no role in designing, conducting and reporting of the study.

**Competing interests:** The authors declared that no competing interests exist.

**Abbreviations:** BP, blood pressure; SBP, systolic blood pressure; DBP, Diastolic blood pressure; CHD, coronary heart disease; DM, diabetes mellitus; CKD, chronic kidney disease; CVD, cardiovascular diseases; IBM, international business machine; JNC-8, eight joint national committees; BB, beta-blocker; CCB, calcium channel blocker; ARB, angiotensin receptor blocker; ACEI, angiotensin-converting enzyme inhibitor; TD, thiazide diuretic; MMAS-8, Morisky medication adherence scale-8; AOR, adjusted odds ratio; COR, crude odds ratio.

= 0.033), and diabetes mellitus co-morbidity (P = 0.008) were significantly associated with a higher rate of medication non-adherence.

## Conclusion

The rate of medication adherence was poor among hypertensive patients. Physicians were not-adherent to standard treatment guideline. The most commonly prescribed class of drugs were angiotensin-converting enzyme inhibitors. Effective education should be given to patients to improve medication adherence. Prescribers should be trained on treatment guidelines regularly to keep them up-to-date with current trends of hypertension treatment and for better treatment outcomes.

## Introduction

Hypertension is a significant public health challenge in the world because of its high prevalence and concomitant risks of cardiovascular and kidney diseases [1]. According to the World Health Organization (WHO), high blood pressure (BP) is a major public health problem that kills one in every eight people and is the world's third-leading silent killer [2].

Almost three-quarters of hypertensive people (639 million people) live in countries with limited health resources [1, 2]. In Sub- Saharan Africa, it is a major independent risk factor for heart failure, stroke, and kidney failure. These complications arise as a result of a low rate of hypertension diagnosis, poor BP control, high morbidity and mortality, and low resources in health care settings [3]. A systematic review and meta-analysis study conducted in Ethiopia in 2015 estimated the prevalence of hypertension to be 19.6% [4]. Another systematic review and meta-analysis conducted in Ethiopia in 2015 found that the prevalence of hypertension ranged from 20% and 30% [5].

Despite the fact that hypertension is a preventable and modifiable risk factor for cardiovascular diseases (CVD), the prevention and control of hypertension have not received attention in many developing countries [6, 7]. It accounts for 40.6% of deaths due to all coronary heart disease and 38.5% of deaths due to stroke [8]. It is also the second-leading cause of end-stage renal disease (ESRD) after diabetes mellitus (DM) [9].

Guideline adherence is defined as a condition in which the prescribed treatment obeys treatments recommended in the identified practice guidelines [10–12]. Hypertension guidelines are evidence-based and are usually dictated by randomized controlled trial data and observational studies. Published guidelines aid in clinical decision making, decrease practice variations, guide correctness, and measure the quality of health care [13, 14]. The Eighth Joint National Committee (JNC-8) on detection, evaluation, and treatment of BP is most the commonly used guideline and it is considered as a "gold standard" consensus guideline for the management of hypertension [15]. This guideline was used as a reference standard for this study because awareness and accessibility of guidelines were initial criteria to evaluate the status of clinicians' adherence to hypertension treatment guidelines. Besides, Ethiopia does not have a hypertension treatment guideline. Therefore, the JNC-8 guideline is widely used by both clinicians and clinical pharmacists in Ethiopia [16]. Clinical practice guidelines do not consistently change clinicians' behavior, control of high blood pressure remains suboptimal, so creating the need to evaluate adherence to guidelines and its impact on BP control [10].

Many studies conducted in the past to explore the causes of suboptimal BP control focused on patient factors such as socio-demographic, medical profile, and patient's treatment

compliance [17–20]. However, prescribing patterns and clinicians' adherence to guidelines are also important factors affecting hypertension treatment outcomes [21–24].

Several studies were conducted worldwide using either prescription or drug dispensing data to evaluate prescription patterns and clinicians' adherence to hypertension treatment guidelines [25–28]. But few data are available in Ethiopia. Also, among previous studies, there were conflicting results between prescribing patterns and patients' medication adherence. A study conducted in Kenya concluded that combination therapies were associated with poor patient adherence [22]. In contrast, in a study conducted in Nigeria combination therapies were associated with good blood pressure controlled than mono-therapy [16]. A study conducted at Jimma University, Ethiopia showed that the number of antihypertensive medications prescribed was not associated with blood pressure control status [18]. Besides, most previous studies used a similar study design(cross-sectional). As a result, a general prospective cohort study was conducted among ambulatory hypertension patients to determine prescribing patterns of antihypertensive drugs, patients' medication adherence, and its determinants.

## Methods

### Study period and setting

This study was conducted over three months among hypertensive patients who were regularly followed up every month at Jimma university ambulatory cardiac clinic from March 20, 2018, to June 20, 2018. Jimma University is the only teaching and specialized hospital in the Jimma zone. It serves over 15 million people living in Jimma city and its surrounding areas. It is located in Jimma town, Oromia Regional State, Ethiopia, 352 km southwest of Addis Ababa, the capital city of Ethiopia [18].

**Study design.**    A general prospective cohort follow-up study was employed.

**Study population.**    All hypertensive patients with age ≥ 18 years, with a documented diagnosis, had at least two times medical appointments before the study period, having a medical appointment every month and complete medical records were included in the study. We excluded pregnant women, patients with confirmed neurologic and psychiatric disorders with their respective specialists since it was challenging to assess medication adherence statuses in acutely ill patients with similar tools like that of other study participants.

**Sample size and sampling technique.**    The sample size for this study was calculated using Cochran single proportion formula. A previous study conducted in Ethiopia reported that the rate of patients' medication adherence was 50% [18]. Based on this formula minimum 384 sample size was calculated using a standard normal distribution (Z = 1.96) with a confidence interval of 95% and a margin of error of 0.05. After adjusting the 10% non-respondent rate, 416 hypertensive patients were included in this study. All clinicians who treated hypertensive patients during the study period were included and a random sampling technique was used to select patients for inclusion in the study.

### Data collection process and data collection tool

Questionnaire and data abstraction format was prepared by reviewing different literatures. To maintain the validity of the data collection tool, a structured questionnaire was developed and translated to local languages (Amharic and Afan Oromo) by native speakers of the respective languages, and then back-translated to the English version. A pre-test of the data collection format was performed on 5% of the sample size. Face-to-face interviews were use to collect primary data from patients, and a self-administered questionnaire was used to collect data from all clinicians who treat hypertensive patients during the study period. During each visit, secondary data such as prescribed drugs, blood pressure measurements, and co-morbid illness

were collected from patients' charts. Initial and subsequent BP is typically measured at Jimma University Medical Center, using the cuff sphygmomanometer and digital sphygmomanometer on the left arm, at the level of the heart, in a sitting position, and at rest for at least 5 minutes. In this study, hypertensive patients who met the inclusion criteria were selected at the first month of the data collection period and BP measurements at this first visit were considered as a baseline. Then only those patients selected at the first month were consequently followed every month for the next three months. Morisky Medication Adherence Scale-8 (MMAS-8) was used to assess the patients' medication adherence. The MMAS-8 total score was calculated by summing the values from all 8 items, with reverse coding when necessary. A score of <2 (out of a full range of 8) was considered as good medication adherence, otherwise considered as poor adherence [19].

Pharmacological Trends Guideline Adherence Evaluation Method was used to measure clinicians' adherence to hypertension treatment guidelines [25]. A recommendation of the JNC-8 on the detection, evaluation, and treatment of high BP was used as a reference guideline to evaluate prescription patterns and clinicians' adherence to the guidelines.

**Data processing and analysis.** The data were entered using Epidata version 3.1 and exported to the Statistical Package for Social Science (SPSS) version 21.0. Continuous variables were presented as means (standard deviation). Categorical variables were presented as frequency and percentages. Antihypertensive medicines were categorized according to their therapeutic classes. Switch from one drug to another drug during each visit, the last regimen was included in the analysis unless changing was done at the last visit.

Guideline adherence was expressed as% = (Total number hypertensive patients treat based on JNC-8 guideline divide by the total number of participants) × 100 [23]. Based on this calculation, >65% conceder as complete adherence, 50 – 64.9% medium adherence, and <49.9% will be classified as low adherence [29]. Finally, BP was calculated by taking an average of three measurements and categorized controlled and uncontrolled based on guidelines. Prior to regression alalysis, a multi-collinearity test was done, and adequacy of cell distribution was checked using the chi-square test. Binary logistic regression was performed to determine the effect of each variable on patients' adherence and variables with a P-value less than 0.25 in the bivariate analysis were then included in a multivariate logistic regression analysis to identify key independent variables influencing patient's medication adherence. In multivariate analysis, variables with P- a value of < 0.05 were considered statistically significant.

**Ethics approval and consent to participates.** This study was approved by the Ethical Review Board of School of Pharmacy, Institute of Health Science, Jimma University (Ref. No IHRPGD/203/18). After a detailed explanation of the objective of the study, procedures of selection, and assurance of confidentiality, each participant were asked orally whether they are voluntary or not to participate in this study. An independent data collection supervisor acted as a witness for voluntary informed decision-making of participants to take part in the study. Written informed consent was waived since the study did not involve any procedure and presented no damage to patients as approved by the ethical review board committee of the Board of School of Pharmacy, Institute of Health Science Jimma University. Their names were not registered to minimize social desirability bias and enhance anonymity. They were not forced to participate or receive any monetary incentive and it was solely voluntary based.

# Results

## Baseline characteristics of study participants

A total of 686 hypertensive patients visited the hypertension clinic during the study period. Four hundred fifty-nine patients fulfilled inclusion criteria, of these, 416 participants were

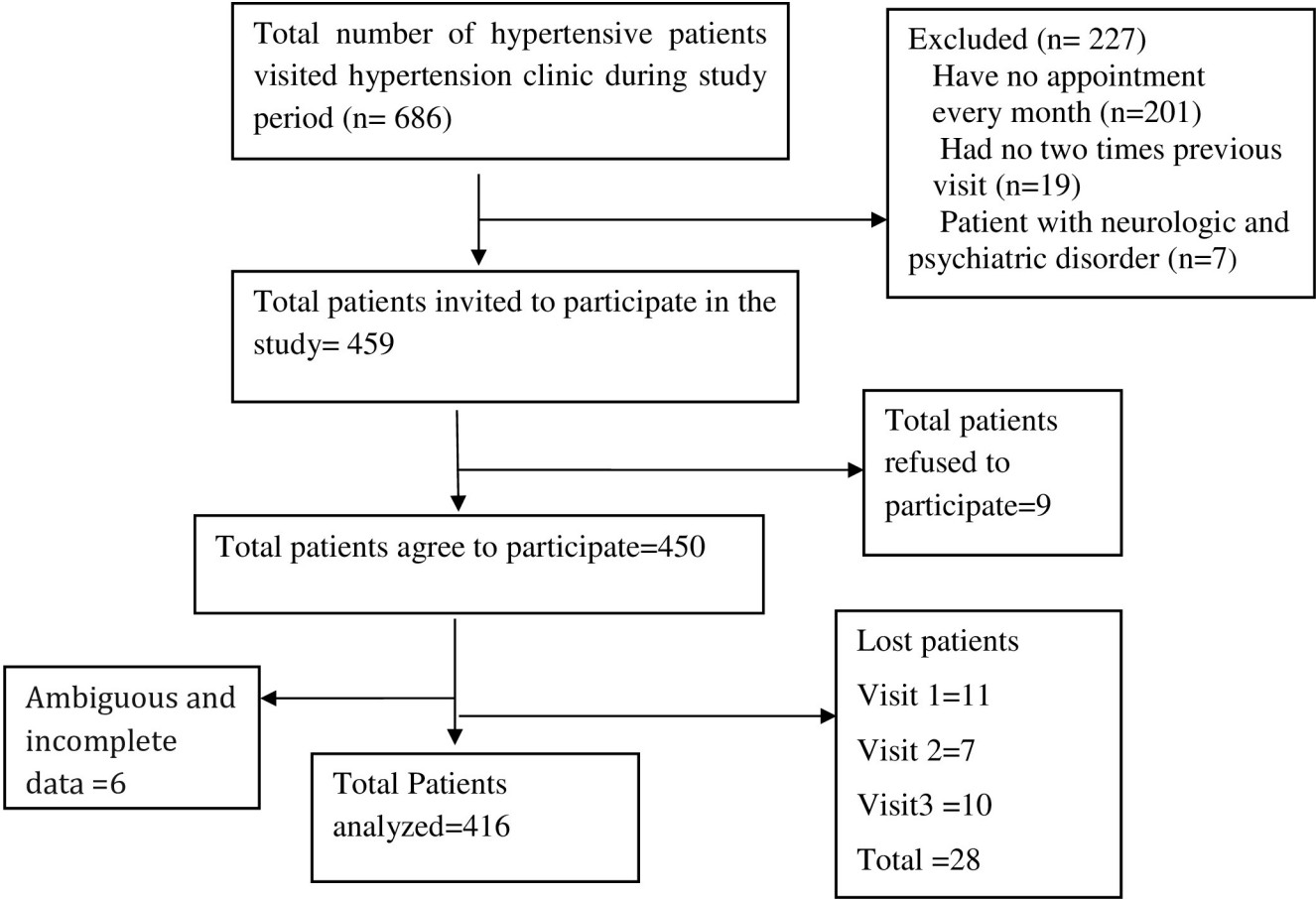

**Fig 1. Hypertension patients selection flow chart at Jimma University Medical Center from March 20, 2018 to June 20, 2018.**

included in the final analyses (Fig 1). More than half (57.0%) of the participants were males with a mean age of 56.50 ± 11.96 years (Table 1).

Two hundred sixty-four (63.5%) of participants have been taking salt with food. More than half (53.4%) of the participants were physically inactive and 138 (33.2%) patients were khat chewers. Among co-morbid conditions, 104(25.0%) of participants had diabetes mellitus (DM), 38(9.1%) and 39(9.4%) had coronary heart disease (CHD) and dyslipidemia, respectively (Table 2).

### Prescription patterns of antihypertensive medications

The overall pattern of antihypertensive agents showed that angiotensin-converting enzyme inhibitor (ACEI) was the most frequently prescribed class of antihypertensive drugs 261 (63.7%), followed by diuretics 234 (57.1%). Among the study participants, 257 (61.7%) of them had one or more concomitant medications, of which 98 (23.6%) and 75(18.0%) of them had two and three drugs concomitant medications, respectively (Table 3).

Of the total participants, 135 (32.5%) were on monotherapy and 275 (66.1%) were on combination therapy. Enalapril was the most frequently prescribed monotherapy 61(14.7%) followed by hydrochlorothiazide 40(9.6%). Among patients who were on combination therapy, two-drug regimens were prescribed in 46.4% of the hypertensive patients. ACEI +diuretics, 75

**Table 1. Socio-demographic characteristics of hypertensive patients at JUMC, 2018.**

| Variables | Characteristics | Frequency (%) |
|---|---|---|
| **Age (year)** | Mean ± SD | 56.50±11.96 |
| | <60 | 220(52.9) |
| | ≥60 | 196(47.1) |
| **Sex** | Male | 237 (57.0) |
| | Female | 179 (43.0) |
| **BMI** | Mean ± SD | 24.24±2.32 |
| | 18.5-24.9 | 314 (75.5) |
| | 25-29.9 | 89 (21.4) |
| | ≥30 | 13 (3.1) |
| **Marital status** | Single | 12 (2.9) |
| | Married | 284 (68.3) |
| | Divorced | 55 (13.2) |
| | Widowed | 65 (15.6) |
| **Residency** | Urban | 274 (65.9) |
| | Rural | 142 (34.1) |
| **Education level** | No formal education | 114 (27.4) |
| | Primary education (1-8 grade) | 110 (26.4) |
| | Secondary education (9-12 grade) | 100 (24.0) |
| | Tertiary education (diploma & above) | 92 (22.1) |
| **Current occupation** | Civil servant | 69 (16.6) |
| | Merchant | 89 (21.4) |
| | Farmer | 74 (17.8) |
| | Housewife | 47 (11.3) |
| | Retired | 36 (8.7) |
| | Jobless | 70 (16.8) |
| | Other | 31 (7.5) |
| **Monthly income (ETB)** | <1000 | 58 (13.9) |
| | 1001-2000 | 68 (16.3) |
| | 2001-3000 | 74 (17.8) |
| | >3000 | 109 (26.2) |
| | Without unknown monthly income | 107 (25.7) |
| **Living status** | Solo | 59 (14.2) |
| | Live with family | 328 (78.8) |
| | Others* | 29 (7.0) |

Others: Drivers, daily laborer, Non-governmental organization and private work; others
*: Live with friends, prison; ETB: Ethiopian Birr.

(18.0%) was a commonly used two-drug combination therapy. Three drugs combination accounted for 17(17.1%) of all prescriptions, of which 35 (8.4%) were on BB+ diuretics+ ACEI followed by CCB+ACEI +diuretic 22(5.3%). Four drugs regimen was prescribed in 11 hypertensive patients, ACEI +diuretics +CCB+BB 7(1.7%) was most frequent prescribed (Table 4).

## Clinicians' adherence to hypertension treatment guidelines

Twenty-five full-time physicians treat hypertensive patients during the study period. There were more males (21) than females (4). The mean age of prescribers was 29.22 ± 4.51 with

**Table 2. Lifestyle factors and clinical characteristics of hypertensive patients at JUMC, 2018.**

| Factors | Categories | Frequency (%) |
|---|---|---|
| Add salt to food | Yes | 264 (63.5) |
| | No | 152 (36.5) |
| Alcohol use | Yes | 103 (24.8) |
| | No | 313 (75.2) |
| Chew khat | Yes | 138 (33.2) |
| | No | 278 (66.8) |
| Cigarette smoking | Never smoked | 303 (72.8) |
| | Ex-smoker | 93 (22.4) |
| | Current smoker | 20 (4.8) |
| Physical activity | Physically active | 194 (46.6) |
| | Physically inactive | 222 (53.4) |
| Drink coffee | Yes | 238 (57.2) |
| | No | 178 (42.8) |
| Use traditional medicine | Yes | 16 (3.8) |
| | No | 400 (96.2) |
| Co morbidity | Hypertension alone | 158 (38.0) |
| | DM | 104 (25.0) |
| | CKD | 16 (3.8) |
| | CHF | 31 (7.5) |
| | CHD | 38 (9.1) |
| | DM and CKD | 16 (3.8) |
| | Dyslipidemia | 39 (9.4) |
| | others | 14 (3.4) |
| Duration on treatment(years) | <1 | 67 (16.1) |
| | 1-5 | 148 (35.6) |
| | >5-10 | 144 (34.6) |
| | >10 | 57 (13.7) |

Others:- Anemia, peripheral neuropathy, lung disease, liver disease, thyroid disorder, and human immunodeficiency virus infection.

range 24-39 years and the majority (17) of them were below 30 years of age. Six of them were general practitioners (GPs), 4 residency year 2 (R2), 8 of them are residency year 3(R3) and 7 of them were internists. Nearly half of clinicians had experience 1-5 years and most of them treating more than 20 hypertensive patients per day. Most prescribers perceived that a JNC-8 hypertension treatment guideline was evidence-based and helpful in the management of patients. However, clinicians' overall adherence to JNC-8 guidelines was low, with only 184 (44.2%) of prescribers following them. From a total of 232 patients not treated based on the JNC-8 guideline, 102(44.0%) patients were not received first-line drugs based on compelling indication and race. In 64(27.6%) patients, combination treatment was not adjusted based on their current BP level (Table 5).

## Patients' adherence to hypertension treatment

A total of 410 study participants received one or more antihypertensive medications. From patients' response to the eight-item Morisky medication adherence Scale, Overall, the prevalence of antihypertensive medication adherence 191 (46.6%)

**Table 3. Frequency of antihypertensive medicines prescribed for hypertensive patients at JUMC, 2018.**

| Classes | Specific drugs | Frequency(%), N = 410 |
|---|---|---|
| ACEIs | Enalapril | 261 (63.7) |
| ARBS | Losartan | 28 (6.8) |
| Diuretics | | 234 (57.1) |
| | Hydrochlorothiazide | 203 (49.5) |
| | Furosemide | 26 (6.4) |
| | Furosemide +spironolactone | 5 (1.2) |
| CCBs | | 144 (35.1) |
| | Amlodipine | 131 (31.9) |
| | Nifedipine | 13 (3.2) |
| BBs | | 105 (25.6) |
| | Atenolol | 45 (10.9) |
| | Metoprolol | 47 (11.5) |
| | Propranolol | 13 (3.2) |
| Co-medications | Total number of patients with co-medication | 257 (61.7) |
| | One drug | 61 (14.7) |
| | Two drugs | 98 (23.6) |
| | Three drugs | 75 (18.0) |
| | ≥ Four drugs | 23 (5.5) |

ACEIs: Angiotensin-converting enzyme inhibitors; ARBs: angiotensin receptor blockers; BBs: βblockers; CCBs: calcium channel blockers.

## Determinants of medication non-adherent among hypertensive patients at JUMC, 2018

In bivariate logistic regression analysis; age, sex, marital status, residency, occupation, monthly income, living status, salt use, khat chewing, physical activity status, concomitant medications, comorbid condition, and duration of treatment were variables with P-value less than 0.25 hence, included in the multivariate logistic regression. However, current occupation, physical activity status, and presence of DM comorbidity were significantly associated with medication non-adherence. Accordingly, merchants were 2.46 times (AOR = 2.46, CI = 1.16-5.23, P = 0.020) more likely to be non-adherent than civil servants. Patients who had no regular physical activity were 1.63 times (AOR = 1.63, CI = 1.04-2.55, P = 0.033) more likely to be non-adherent compared to physically active patients. Hypertensive patients with DM comorbidity were 2.54 times (AOR = 2.54, CI = 1.28-5.04, P = 0.008) more likely to be non-adherent as compared to clients with no comorbidity (Table 6).

## Discussion

Medication adherence is the main predictor of treatment success and an important step in lowering BP. The present study showed that the overall rate of patients' medication adherence was 191 (46.6%). The finding was almost similar to the WHO report in 2011 in developing countries (50%) [30]. It was also comparable to studies done in Egypt (46.12%) and Saudi Arabia (46%) [31, 32]. It was, however, lower than studies reported from Taiwan (53%), Sweden (63.1%), and China (53.4%) [33–35]. The adherence discrepancy between the studies could be due to variations in the studied populations, better health care, and access to health in the developed countries. Moreover, differences in study results could be due to differences in measurement methodologies employed to assess drug adherence. For example, in Taiwan medication adherence was measured using the medication possession ratio (percentage of time that

**Table 4. Regimens of antihypertensive therapy among hypertensive patients at JUMC, 2018.**

| Regimens | Specific regimens | Frequency (%) |
|---|---|---|
| Non-pharmacologic | | 6 (1.4) |
| Monotherapy | | 135 (32.5) |
| | Enalapril | 61 (14.7) |
| | Amlodipine | 26 (6.2) |
| | Nifedipine | 4 (1.0) |
| | Hydrochlorothiazide | 40 (9.6) |
| | Losartan | 4 (1.0) |
| Dual therapy | | 193 (46.4) |
| | ACEI + Diuretic | 75 (18.0) |
| | CCB + BB | 6 (1.4) |
| | Diuretic + BB | 16 (3.8) |
| | ACEI + CCB | 30 (7.2) |
| | ACEI + BB | 24 (5.8) |
| | Diuretic +ARB | 4 (1.0) |
| | CCB + ARB | 8 (1.9) |
| | CCB+ Diuretics | 26 (6.3) |
| | ARB+BB | 4 (1.0) |
| Triple therapy | | 71 (17.1) |
| | BB + Diuretic + ACEI | 35 (8.4) |
| | CCB + BB + Diuretic | 7 (1.7) |
| | CCB +ACEI + Diuretic | 22 (5.3) |
| | CCB + BB + ACEI | 1 (0.2) |
| | CCB + BB +ARB | 3 (0.7) |
| | Others* | 3 (0.7) |
| Quadruple | | 11 (2.6) |
| | ACEI + Diuretics + CCB + BB | 7 (1.7) |
| | ARB + BB + CCB + diuretic | 1 (0.2) |
| | Others** | 3 (0.7) |

Others

*: Enalapril +Furosemide +Spironolactone, Amlodipine +Furosemide + Hydrochlorothiazide; others

**: Atenolol+ Enalapril+ Furosemide+ Spironolactone, Amlodipine+ Enalapril+ Furosemide+ Spironolactone.

the patient had medication available to them during the follow-up period) whereas, in Sweden adherence was measured by using the Proportion of Days Covered (PDC) method.

This finding was also lower than the studies conducted in Gondar and Jimma university (64.6% and 61.8%, respectively) [36, 37]. The discrepancy might be due to the varaition in the inclusion criteria and adherence measurement scale. Participants in the Jimma University

**Table 5. Common problems observed from prescribed antihypertensive medications to hypertensive patients at JUMC concerning the recommendation of JNC-8 guideline, N = 232.**

| Compliance issue | Frequency | Percent |
|---|---|---|
| Patients did not receive first-line drugs based on compelling indication and race recommendation | 102 | 44.0 |
| Combination treatment was not adjusted based on their current blood pressure level | 64 | 27.6 |
| Patients received an inappropriate dose of medications | 50 | 21.6 |
| Patients were not on first-line drugs and the right dose of medication | 16 | 6.9 |

**Table 6. Results of logistic regression analysis for factors associated with medication non-adherence among adult hypertensive patients at JUMC, 2018.**

| Variables | | Medication adherence | | COR (95%CI) N = 410 | P-value | AOR (95%CI) | P- value |
|---|---|---|---|---|---|---|---|
| | | Adherent | Non Adherent | | | | |
| Occupation | Civil servant | 40 | 27 | 1.00 | 1.000 | 1.00 | 1.000 |
| | Merchant | 30 | 58 | 2.86(1.48-5.53) | 0.002 | 2.46(1.16-5.23) | 0.020 |
| | Farmer | 29 | 43 | 2.20(1.12-4.33) | 0.023 | 1.41(0.52-3.78) | 0.501 |
| | House wife | 24 | 22 | 36(0.64-2.89) | 0.428 | 12(0.43-2.91) | 0.812 |
| | Retired | 12 | 24 | 66(1.27-6.92) | 0.012 | 04(0.66-6.26) | 0.215 |
| | Jobless | 38 | 32 | 1.25(0.63-2.46) | 0.522 | 0.71(0.26-1.91) | 0.500 |
| | Others | 18 | 13 | 1.07(0.45-2.54) | 0.878 | 0.73(0.26-2.08) | 0.560 |
| Physical activity | Inactive | 89 | 130 | 1.67(1.13-2.48) | 0.010 | 1.63(1.04-2.55) | 0.033 |
| | Active | 102 | 89 | 1.00 | 1.000 | 1.00 | 1.000 |
| Comorbidities | HTN alone | 84 | 72 | 1.00 | 1.000 | 1.00 | 1.000 |
| | DM | 37 | 64 | 2.02(1.21-3.37) | 0.007 | 2.54(1.28-5.04) | 0.008 |
| | CKD | 11 | 5 | 53(0.18-1.59) | 0.260 | 51(0.15-1.71) | 0.272 |
| | CHF | 14 | 17 | 42(0.65-3.01) | 0.378 | 14(0.48-2.71) | 0.763 |
| | CHD | 13 | 25 | 24(1.10-4.70) | 0.032 | 13(0.88-5.14) | 0.093 |
| | DM+CKD | 11 | 5 | 53(0.18-1.60) | 0.260 | 51(0.15-1.79) | 0.294 |
| | Dyslipidemia | 17 | 21 | 44(0.71-2.94) | 0.315 | 14(0.51-2.53) | 0.751 |
| | Others** | 4 | 10 | 92(0.88-9.69) | 0.081 | 81(0.77-10.22) | 0.116 |

COR = crude odds ratio; AOR = Adjusted odds ratio: others

**: Drivers, daily laborer, non-governmental organization and private work; others**: Anemia, peripheral neuropathy, lung disease, liver disease, thyroid disorder, and human immunodeficiency virus infection.

study were hypertension patients who had been followed for at least a year, and the sample size (280 patients) was lower than in this study. The Morisky 4-item Medication Adherence Scale was used to assess adherence at Gondar University. The medication adherence level found in this study, on the other hand, was higher than the finding from Ghana and Nigeria (30.3%) and Iran (24%) [38, 39]. This disparity could be explained by the study done in Ghana and Nigeria included hypertensive patients who had depression which may have contributed to the low medication adherence. Similarly, in Iran, participants were selected from the rural area only which might have contributed to medication non-adherence.

In this study, a significant association was observed between patients' current occupation and medication nonadherence. Merchant hypertensive patients were 2.46 times more likely to be nonadherent than civil servants. This finding was in line with studies done in Hong Kong China and Black Loin Specialized Hospital, Ethiopia [40, 41]. This might be because they forget to take drugs with themselves when they travel or leave home. Besides, they may be too busy to come to the health facility for their pills, and they may have difficulties in remembering to take all of their medicine on time. Physical inactivity was another important factor influencing patients' adherence in this study. Patients who had no regular physical activity were 1.63 times more likely to be non-adherent than physically active patients. This was consistent with a finding from Iran [39]. The exact mechanism was unclear, but it might be that physically inactive hypertensive patients will have uncontrolled BP (57.2% in this study), prompting clinicians to prescribe a more complex treatment regimen. Furthermore, patients with uncontrolled BP might be hopeless to take their medications and adhere to their treatment plan.

In addition, the presence of DM as comorbidity was significantly associated with poor antihypertensive medication adherence. Hypertensive patients with DM comorbidity were 2.54

times more likely to be nonadherent than with no co-morbidities. This finding was similar to studies conducted in South Korea, Hong Kong China, Saudi Arabia, and Gondar, Ethiopia [10, 32, 36, 40]. This could be explained by patients with co-morbidities who may have suffered from serious complications and had complex treatment regimens making it difficult for them to be adherent to their medications. Furthermore, 27.4% of participants in this study had no formal education, making it difficult to adhere to treatment plans and take complex treatment on time. This indicated that close supervision, short time appointments and fixed-dose combinations are important to ehance medication adherence for this group of patients.

This study found that 44.2% of clinicians' were adherent to hypertension treatment guidelines. The rate of adherence was slightly higher than the study conducted in Zewditu Memorial Hospital, Ethiopia (37.4%) [26]. This difference might be due to the small sample size, and most of the study participants (65.2%) were without co-morbid conditions in Zewditu Memorial Hospital. However, it was lower than studies conducted in South Indian (65%), Malaysia (85.30%), Island (70.4%), and South Africa (51.9%) [21, 23, 29, 42]. This disparity might be due to racial differences, as well as lack of laboratory and imaging facilities in our setting for further screening of target organ damage. In addition, it might be explained by the difference in prescribers profiles since the majority of doctors treating hypertensive patients in this study were general practitioners and residents. Six of them were general practitioners (GP), 4 residency year 2 (R2), 8 of them are residency year 3(R3) from the total of 25 doctors. The common challenges to adherence to JNC-8 hypertension treatment guidelines reported by prescribers were the cost of medication, lack of medication availability, and the presence of comorbidity. Ethiopias' Ministry of Health and other concerned bodies must continue to prioritize health care accessibility and affordability. The strong commitment of the Ethiopian ministry of health and the institution for health care improvement outcomes including fidelity, accesseblity of well-trained health care professionals, laboratory and imaging facilities for further screening for target organ damage, and program effectiveness are important to enhance clinicians guideline adherence in a health care setting.

According to JNC-8 guideline, CCBs and thiazide diuretics are the recommended first-line medicines for the management of hypertension for Africa origin. However, ACEIs were prescribed first, followed by diuretics and CCBs in this study (63.7%, 57.1%, 35.1% respectively). The most commonly prescribed class of drug among hypertensive patients with DM, CHF, or CHD was ACEI (66.3 percent, 71.0 percent, and 65.8 percent, respectively), whereas diuretics were the most commonly prescribed drug in CKD. This might be because prescribers (clinicians) were not adherent to standard treatment guidelines (JNC-8). ACEIs are preferable for CKD patients due to renal protective effects. The overall use of ACEIs was comparable to studies conducted in Serbia (60.57%) and Mexico City (63.8%) [43, 44]. Thiazide diuretics were the most frequently prescribed medications in studies conducted in the Eastern Central Region of Portugal and rural tertiary hospitals in Nigeria showed that (67%, and 84.9%), respectively [16, 29]. This disparity could be attributed to studies in Portugal and Nigeria that used JNC-7 as a reference guideline, which is conservative in its use of thiazide-type diuretics as first-line therapy for most patients unless there is a compelling indication.

The result of this study showed that 135 (32.5%) patients were on mono-therapy regardless of the presence or absence of comorbidities, which was lower than the studies reported from Mexico (72.1%), Canada (56.3%), and Turkey (75.7%) [44–46]. The difference might be due to better health care, availability of medications, and close monitoring in these countries which help to achieve target BP with a single medication. Of the total of patients who were on monotherapy, ACEIs, 61 (14.7%) were the most prescribed drug classes. The result was closely similar to a study conducted in Kenya (20.2%) [47]. However, it was lower than the study conducted in Turkey (30.1%) [46]. The variation might be due to using different standard

reference guidelines in which European guidelines recommend any class of drug as initial therapy. On contrary, the study conducted in Gondar hospital showed that thiazide diuretics were the most commonly prescribed mono-therapy (60.24%) [48]. this discrepancy might be because of the difference in level guideline adherence (66.8% prescription based on JNC-8 guidelines) that recommended the use of diuretics for both mono and combination therapy. In addition, variation may be different in the prevalence of CKD co-morbidity, as only 3.8% of participants had CKD comorbidity in this study.

On the other hand, the majority of the patients, 275(66.1%) were on combination therapy. The finding was consistent with the study conducted in Kenya (60%) [47]. It was also similar to a study conducted at Zewditu Memorial Hospital Ethiopia (70.8%) [26]. However, the finding of this study was higher than the study conducted in Malaysia (56.7%) [49]. The higher prescription rate of combination therapy in this study might be due to the low rate of BP control 42.8% in this study as compared to 84.6% in Malaysia.

Two drugs regimen accounted for 46.4% of the combination therapy, of this 75(18%) were ACEIS + diuretics. The finding was in line with a study conducted in Kenya (14.5%) [47]. A study conducted in India using the JNC-8 guideline showed that the most frequently prescribed two-drug combination was ARB +diuretics [25]. This variation might be due to the cost and easy accessibility of ARB in India. On the other hand, in the study conducted in Nigeria, CCB + diuretic was the most frequently used two-drug combination (36.6%) [16]. This variation might be suggesting of doctor's preference to use the CCBs combination as initial therapy. Besides, in Nigeria majority of the patients were elderly (mean age was 61.5±15.1 years) hence, CCBs are preferable for older patients because isolated SBP is more prevalent due to vessel stiffness [50].

## Limitations of the study

This study has some limitations. The study was conducted in one facility, therefore; the findings may not be generalized to reflect all health care setting in Ethiopia. Only prescription and co-morbidity data were used to examine compliance to treatment guidelines, which may not always be reliable. Finally, this study was unable to identify factors affecting clinicians' adherence to standard treatment guidelines but will be an interesting area for future research.

## Conclusion

The rate of medication adherence was poor among hypertensive patients. Being merchant, physical inactivity, and the presence of DM co-morbidity were factors associated with poor medication adherence. Prescribing patterns of antihypertensive drugs were inconsistent with JNC-8 guidelines. Angiotensin-converting enzyme inhibitors were the most frequently prescribed class of anti-hypertensive drugs. The majority of participants were on combination therapy. Effective education should be given to patients to improve medication adherence. Prescribers should be trained on treatment guidelines regularly to keep them up-to-date with current trends of hypertension treatment and for better treatment outcomes.

## Supporting information

**S1 Data.**
(SAV)

**S1 File.**
(DOCX)

## Acknowledgments

We thank the participants of the study.

## Author Contributions

**Conceptualization:** Bekalu Kebede Simegn, Legese Chelkeba.

**Data curation:** Bekalu Kebede Simegn.

**Formal analysis:** Bekalu Kebede Simegn, Legese Chelkeba.

**Investigation:** Bekalu Kebede Simegn.

**Methodology:** Bekalu Kebede Simegn.

**Supervision:** Legese Chelkeba.

**Visualization:** Bekalu Kebede Simegn, Bekalu Dessie Alamirew.

**Writing – original draft:** Bekalu Kebede Simegn, Legese Chelkeba, Bekalu Dessie Alamirew.

**Writing – review & editing:** Bekalu Kebede Simegn, Bekalu Dessie Alamirew.

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
