## [Decision Letter · Decision Letter 0]

8 Apr 2021

PONE-D-20-38133

Clinician’s Prescribing Pattern, Rate of Patients’ Medication Adherence and Its Determinants Among Adult Hypertensive   Patients in Ethiopia:  Prospective Cohort Study

PLOS ONE

Dear Dr. simegn,

Thank you for submitting your manuscript to PLOS ONE. After careful consideration, we feel that it has merit but does not fully meet PLOS ONE’s publication criteria as it currently stands. Therefore, we invite you to submit a revised version of the manuscript that addresses the points raised during the review process.

We look forward to receiving your revised manuscript.

Kind regards,

Pathiyil Ravi Shankar

Academic Editor

PLOS ONE

Journal Requirements:

2. Please include additional information regarding the survey or questionnaire used in the study and ensure that you have provided sufficient details that others could replicate the analyses. For instance, if you developed a questionnaire as part of this study and it is not under a copyright more restrictive than CC-BY, please include a copy, in both the original language and English, as Supporting Information. Moreover, please include more details on how the questionnaire was pre-tested, and whether it was validated.

3. Please provide additional details regarding participant consent. In the ethics statement in the Methods and online submission information, please ensure that you have specified how verbal consent was documented and witnessed.

4. We suggest you thoroughly copyedit your manuscript for language usage, spelling, and grammar. If you do not know anyone who can help you do this, you may wish to consider employing a professional scientific editing service.  

'This research was funded by Jimma University, College of Health Sciences.

The University has no role in designing, conducting and reporting of the study.'

'The funders had no role in study design, data collection and analysis, decision to

publish, or preparation of the manuscript.'

Additional Editor Comments (if provided):

Dear authors

Thank you for submitting your manuscript to PLoS One. Your manuscript has been peer-reviewed by three expertis in the field. They have pointed out a number of problems with youre manuscript including the fact that in its present form the manuscript is difficult tro understand and has not been presented in a proper format. My onw reading of the manuscript corroborates their opinion.

I understand you work in difficult circumstances and I have decided to provide you with an option to completely revise and resubmit your manuscript. The resubmitted manuscript will again be peer reviewed. If you are able to revise sufficiently to satisfy the concerns of the reviewers and my own concerns we will move further. Also deposit the data connceted with your research in a publicly avaialble data repository after removing identying patient details if you have not already done so. Otherwise we will not proceed further with this submission.

Thank you once again and looking forward to a revised submission

With regards

Dr Shankar

Reviewers' comments:

Reviewer's Responses to Questions

**Comments to the Author**

1. Is the manuscript technically sound, and do the data support the conclusions?

Reviewer #1: No

Reviewer #2: Partly

Reviewer #3: Yes

2. Has the statistical analysis been performed appropriately and rigorously? 

Reviewer #1: I Don't Know

Reviewer #2: Yes

Reviewer #3: Yes

3. Have the authors made all data underlying the findings in their manuscript fully available?

Reviewer #1: No

Reviewer #2: Yes

Reviewer #3: Yes

4. Is the manuscript presented in an intelligible fashion and written in standard English?

Reviewer #1: No

Reviewer #2: No

Reviewer #3: No

5. Review Comments to the Author

Reviewer #1: Authors need to rework on the manuscript for better clarity. The language is not clear. The methodology is also not robust. The number of tables are also more with lengthy descriptions. Discussion section also requires rewriting and conclusion has to be based upon the findings and the possible recommendations.

The comments and suggestions are given in the reviewed manuscript.

Reviewer #2: Abstract

"March 20 to June 20, 2018", please make this clearer. March 20 of which year?

Introduction

There are a lot of issues here. For example, you mentioned on page 3 that "Ethiopia has not its own hypertension treatment guideline and on the time being this guideline widely using by both clinicians and clinical pharmacist [16]". What exactly do you mean? The sentence is confusing. I suggest you get someone to help with the arrangement of your sentences.

Method

Please what do you mean by patients were consequently followed for the next three months? Was done done daily, weekly, monthly, etc for all participants (page 5)?

Your sample size was calculated based on patients' adherence to treatment, do you think that this will adequately cater for physicians adherence to guidelines?

Page 5, please what do you mean by "questioner"?

Please check your definition of Guideline adherence. I think the denominator "total number of hypertensive patients treated". I also think that the Operational Definition should be at the Appendix and referred to periodically.

Page 5, please how were your BP measurements taken?

Results

Your tables, in particular table 6 should be reformatted. It is very difficult to following the distributions.

Table 3, please why do you put those with one medication under co-medication? Enalapril constitutes 63.7% of ACEIs, where are the rest? Same with ARBs. Why have you presented ACEIs and ARBs different from Diuretics, CCBs and BBs?

Table 6 is not clear at all. I could not read this well, but there are issues with it.

Conclusion

Because most of your information are on Table 6 and I could not read this well, I did not review your discussion.

Reviewer #3: There are many grammatical errors in the manuscript that tend to lose the reader. They have been included in the reviewed manuscript uploaded. The manuscript looks like it has been copy pasted from a dissertation/thesis; only key information should be included and the rest uploaded. These sections have been highlighted in the attached manuscript. There are very minimal results on the clinicians' adherence to treatment guidelines and these results are not discussed, yet they seem to form a huge part of this manuscript. In the discussion section, the authors should discuss only the key findings and not every finding and include the implications of such findings. This has been scantily done. There are too many tables and figures in the manuscript- the authors should include only key findings and present the rest as attachments. The authors should review author guidelines and revise the manuscript accordingly.

6. PLOS authors have the option to publish the peer review history of their article (what does this mean?). If published, this will include your full peer review and any attached files.

Reviewer #1: No

Reviewer #2: **Yes: **Daniel N A Ankrah

Reviewer #3: **Yes: **Sylvia Opanga

---

## [Author Response · Author response to Decision Letter 0]

15 Jul 2021

First, we would like to appreciate the editor and reviewers for giving us another round of invaluable comments so as to revise our manuscript accordingly. We have gone through all the comments given by the reviewers and revised the manuscript point by point.

---

## [Decision Letter · Decision Letter 1]

28 Jul 2021

PONE-D-20-38133R1

Clinicians’ Prescribing Pattern, Rate of Patients’ Medication Adherence and Its Determinants among Adult Hypertensive Patients at Jimma University Medical Center:  Prospective Cohort Study

PLOS ONE

Dear Dr. simegn,

Thank you for submitting your manuscript to PLOS ONE. After careful consideration, we feel that it has merit but does not fully meet PLOS ONE’s publication criteria as it currently stands. Therefore, we invite you to submit a revised version of the manuscript that addresses the points raised during the review process.

The reviewers' comments are appended. There are problems with English usage which interfere with the readability of the manuscript. I request you to kindly get your manuscript copyedited for language and gramamr either by a native English speaker or a copyediting service and attach the letter of proof reading with the revised submission. I can empathize with you as you are not a native English speaker but languagae corrections are required before the manuscript can be considered for publication. 

We look forward to receiving your revised manuscript.

Kind regards,

Pathiyil Ravi Shankar

Academic Editor

PLOS ONE

Journal Requirements:

Reviewers' comments:

Reviewer's Responses to Questions

**Comments to the Author**

1. If the authors have adequately addressed your comments raised in a previous round of review and you feel that this manuscript is now acceptable for publication, you may indicate that here to bypass the “Comments to the Author” section, enter your conflict of interest statement in the “Confidential to Editor” section, and submit your "Accept" recommendation.

Reviewer #1: All comments have been addressed

Reviewer #2: (No Response)

Reviewer #3: (No Response)

2. Is the manuscript technically sound, and do the data support the conclusions?

Reviewer #1: Yes

Reviewer #2: Yes

Reviewer #3: Yes

3. Has the statistical analysis been performed appropriately and rigorously? 

Reviewer #1: I Don't Know

Reviewer #2: Yes

Reviewer #3: Yes

4. Have the authors made all data underlying the findings in their manuscript fully available?

Reviewer #1: Yes

Reviewer #2: Yes

Reviewer #3: Yes

5. Is the manuscript presented in an intelligible fashion and written in standard English?

Reviewer #1: (No Response)

Reviewer #2: No

Reviewer #3: Yes

6. Review Comments to the Author

Reviewer #1: Authors have answered to all the comments and queries given to them. The manuscript is now in a better shape.

Reviewer #2: I can still see the statements that do not sound so interesting. For example on page 2 the writers said, “Besides, Ethiopia has not its hypertension treatment guideline”. This can be captured as “Besides, Ethiopia does not have a treatment guideline for hypertension”.

Page 18, there is nothing under Declaration. This should have a statement on conflict of interest.

I strongly feel you need someone to read and correct some of the phrases in your manuscript.

Reviewer #3: There are still a few typographical and grammatical errors that the authors need to address. These have been highlighted in the comment section of the manuscript.

The discussion section needs to be beefed up- it should include a discussion of the implications of the findings on policy and practice, and not a mere comparison of findings with other studies in literature. Suggestions have been made in the comments section of the manuscript. Only the key findings should be discussed; as opposed to what the authors have done, discussing every finding. Tables and figures should be reduced- authors should see which ones to remove (reviewer has suggested one) so that only the prescribed number of tables is included , with the rest provided in the supplementary material section.

Plesae see attached review.

7. PLOS authors have the option to publish the peer review history of their article (what does this mean?). If published, this will include your full peer review and any attached files.

Reviewer #1: No

Reviewer #2: **Yes: **Daniel Nii Amoo Ankrah

Reviewer #3: **Yes: **Dr Sylvia Opanga

---

## [Author Response · Author response to Decision Letter 1]

20 Aug 2021

we have tried to address all comments given by reviewers

---

## [Decision Letter · Decision Letter 2]

20 Oct 2021

Clinicians’ Prescribing Pattern, Rate of Patients’ Medication Adherence and Its Determinants among Adult Hypertensive Patients at Jimma University Medical Center:  Prospective Cohort Study

PONE-D-20-38133R2

Dear Dr. simegn,

We’re pleased to inform you that your manuscript has been judged scientifically suitable for publication and will be formally accepted for publication once it meets all outstanding technical requirements.

Kind regards,

Pathiyil Ravi Shankar

Academic Editor

PLOS ONE

Additional Editor Comments (optional):

Reviewers' comments:

Reviewer's Responses to Questions

**Comments to the Author**

1. If the authors have adequately addressed your comments raised in a previous round of review and you feel that this manuscript is now acceptable for publication, you may indicate that here to bypass the “Comments to the Author” section, enter your conflict of interest statement in the “Confidential to Editor” section, and submit your "Accept" recommendation.

Reviewer #2: All comments have been addressed

2. Is the manuscript technically sound, and do the data support the conclusions?

Reviewer #2: Yes

3. Has the statistical analysis been performed appropriately and rigorously? 

Reviewer #2: Yes

4. Have the authors made all data underlying the findings in their manuscript fully available?

Reviewer #2: Yes

5. Is the manuscript presented in an intelligible fashion and written in standard English?

Reviewer #2: Yes

6. Review Comments to the Author

Reviewer #2: I have no comments to make at this moment. In my view, the authors have tried hard to meet all the comments I raised earlier.

7. PLOS authors have the option to publish the peer review history of their article (what does this mean?). If published, this will include your full peer review and any attached files.

Reviewer #2: **Yes: **Daniel Nii Amoo Ankrah

---

## [Editor Report · Acceptance letter]

5 Nov 2021

PONE-D-20-38133R2 

Clinicians’ Prescribing Pattern, Rate of Patients’ Medication Adherence and Its Determinants among Adult Hypertensive Patients at Jimma University Medical Center:  Prospective Cohort Study 

Dear Dr. Simegn:

I'm pleased to inform you that your manuscript has been deemed suitable for publication in PLOS ONE. Congratulations! Your manuscript is now with our production department. 

Kind regards, 

on behalf of

Dr. Pathiyil Ravi Shankar 

Academic Editor

PLOS ONE